# Optimizing Micropropagation and Rooting Protocols for Diverse Lavender Genotypes: A Synergistic Approach Integrating Machine Learning Techniques

Özhan Şimşek [1,*], Akife Dalda Şekerci [1], Musab A. Isak [2], Fatma Bulut [1], Tolga İzgü [3], Mehmet Tütüncü [4] and Dicle Dönmez [5]

1   Horticulture Department, Agriculture Faculty, Erciyes University, 38030 Kayseri, Türkiye; akifedalda@erciyes.edu.tr (A.D.Ş.); fatmabulut9434@gmail.com (F.B.)
2   Agricultural Sciences and Technology Department, Graduate School of Natural and Applied Sciences, Erciyes University, 38030 Kayseri, Türkiye; musabisak11@gmail.com
3   Institute of BioEconomy, National Research Council of Italy (CNR), 50019 Florence, Italy; tolga.izgu@ibe.cnr.it
4   Department of Horticulture, University of Ondokuz Mayıs, 55200 Samsun, Türkiye; mehmet.tutuncu@omu.edu.tr
5   Biotechnology Research and Application Center, Çukurova University, 01330 Adana, Türkiye; dicledonmez4@gmail.com
*   Correspondence: ozhansimsek@erciyes.edu.tr

**Abstract:** This study comprehensively explored the micropropagation and rooting capabilities of four distinct lavender genotypes, utilizing culture media with and without 2 g/L of activated charcoal. A systematic examination of varying concentrations of BAP for micropropagation and IBA for rooting identified an optimal concentration of 1 mg/L for both BAP and IBA, resulting in excellent outcomes. Following robust root development, the acclimatization of plants to external conditions achieved a 100% survival rate across all genotypes. In addition to the conventional techniques employed, integrating machine learning (ML) methodologies holds promise for further enhancing the efficiency of lavender propagation protocols. Using cutting-edge computational tools, including MLP, RBF, XGBoost, and GP algorithms, our findings were rigorously examined and forecast using three performance measures (*RMSE*, $R^2$, and *MAE*). Notably, the comparative evaluation of different machine learning models revealed distinct R2 rates for plant characteristics, with MLP, RBF, XGBoost, and GP demonstrating varying degrees of effectiveness. Future studies may leverage ML models, such as XGBoost, MLP, RBF, and GP, to fine-tune specific variables, including culture media composition and growth regulator treatments. The adaptability and ability of ML techniques to analyze complex biological processes can provide valuable insights into optimizing lavender micropropagation on a broader scale. This collaborative approach, combining traditional in vitro techniques with machine learning, validates the success of current micropropagation and rooting protocols and paves the way for continuous improvement. By embracing ML in lavender propagation studies, researchers can contribute to advancing sustainable and efficient plant propagation techniques, thereby fostering the preservation and exploitation of genetic resources for conservation and agriculture.

**Keywords:** micropropagation; rooting efficiency; activated carbon; BAP; IBA

## 1. Introduction

Lavender (*Lavandula* L.), a valuable aromatic plant from the Lamiaceae family [1], is economically significant as an essential oil and attractive ornamental plant in arid regions. *Lavandula*, a genus widely distributed from North Africa to the Mediterranean, southwestern Asia, Arabia, western Iran, and eastern India, comprises over 39 known species [2]. Lavender cultivation is practiced in many countries, including Argentina, Brazil, Bulgaria, Cyprus, Greece, Croatia, Hungary, Iran, Italy, Russia, Spain, Türkiye, Japan, and Great Britain [3]. France, Bulgaria, England, the USA, North Africa, and Türkiye

also economically cultivate lavender. Mainly, *L. hybrida* and *L. angustifolia* are species of economic importance due to their quantity and quality of essential oils.

The essential oils from these species find applications in the perfume and fragrance industry. Some are widely used in aromatherapy and are known for their antiseptic and antimicrobial properties [4].

The composition and yield of oil in the *Lavandula* species distinguish them. Standard criteria for determining oil quality include the proportions of camphor, linalool, and linalyl acetate in the essential oil [5]. In recent years, increased interest in lavender cultivation has heightened the significance of scientific research on advanced cultivation techniques, leading to significant developments in this field. The propagation of lavender has become a critical aspect of lavender production, gaining considerable importance worldwide. Lavender propagation can be achieved through two main methods: generative and vegetative. Some *Lavandula* species can only be propagated generatively through their seeds, while others can be propagated vegetatively through stem cuttings. Certain lavender species and varieties can be efficiently propagated using both methods, offering a quicker and more accessible means of reproduction. *L. angustifolia* and *L. spica*, with a diploid (2x = 2n) and tetraploid (4x = 2n) structure, are suitable for both generative and vegetative propagation. In contrast, triploid (3x = 2n) variations of *Lavandula* not L. are sterile and cannot produce seeds; they can only be propagated vegetatively [6]. The propagation performance of different *Lavandula* species and genotypes varies. Similar to many plant species, the propagation of lavender genotypes through plant tissue culture has become an essential and advantageous practice.

Plant tissue culture is an important technique across various domains of research and practical applications. It encompasses the cultivation and development of plant cells, tissues, or organs within a controlled environment facilitated by an artificial nutrient medium [7]. This method has brought about a paradigm shift in plant biotechnology, serving as a crucial instrument for producing high-quality plant-based medicines, consistently generating biologically active compounds and conserving endangered plant species [8].

In summary, plant tissue culture is a pivotal technique that has many applications and advantages. It enables the production of genetically uniform and disease-free plant material, the consistent generation of biologically active compounds, the preservation of endangered plant species, and the exploration of plant biology. Despite these limitations, tissue culture has ushered in a revolution in plant biotechnology and continues to be an indispensable tool for researchers and practitioners.

Micropropagation, also known as in vitro propagation, plays a pivotal role in the large-scale production, conservation, and enhancement of lavender plants. This technique involves cultivation and development of lavender cells, tissues, or organs under controlled conditions in an artificial nutrient medium [9]. The composition of the culture medium holds paramount importance in lavender micropropagation, as it dictates the plant tissue's growth, morphological attributes, and phytochemical makeup [10]. Numerous studies have been dedicated to refining micropropagation protocols for lavender, revealing that shoot proliferation increases in initial subcultures but declines in subsequent ones [11,12]. Elicitor compounds such as jasmonic and salicylic acid, integrated into the culture medium, have positively impacted the growth and biochemical composition of in vitro propagated lavender [10]. These elicitors effectively enhance the production of secondary metabolites, including the essential oils distinctive to lavender, which find versatile applications in the pharmaceutical, cosmetic, and food industries [13]. Meristem culture is employed in lavender micropropagation, utilizing apical meristem or axillary buds as explants for initiation [14]. Shoot multiplication is achieved by forming adventitious roots [9], which is a crucial step in facilitating the successful acclimatization and establishment of micropropagated lavender plants under field conditions [15]. Incorporating calcium into the culture medium has been instrumental in addressing issues such as hyperhydricity and shoot-tip necrosis, familiar challenges encountered during lavender micropropagation [16].

In addition to protocol optimization, research endeavors have aimed to characterize micropropagated lavender plants. Studies have revealed distinctions in the essential oil composition between in vitro propagated lavender and field-grown counterparts, contributing to heightened antioxidant and antimicrobial activities in micropropagated plants. These essential oils have also been scrutinized for potential applications in cosmetic formulations [13]. Furthermore, micropropagation techniques offer a means for preserving and conserving lavender germplasm. As lavender remains a sought-after ornamental crop, efficient in vitro propagation methods are essential to mitigate the overexploitation of natural populations [17,18]. Micropropagation facilitates the rapid multiplication and preservation of elite lavender cultivars and presents opportunities for creating novel forms and establishing clonal micropropagation systems [19]. In summation, micropropagation emerges as a valuable tool in the mass production, conservation, and enhancement of lavender plants. The refinement of culture media, the application of elicitors, and the comprehensive characterization of micropropagated plants collectively contribute to the successful integration of this technique into lavender production. Micropropagation presents a sustainable and efficient means of propagating lavender, ensuring a consistent supply of high-quality plants for diverse industries and safeguarding lavender germplasm for future generations.

Machine learning (ML) represents the application of data science techniques to address intricate challenges across various scientific domains. However, the utilization of ML methodologies in the context of plant and agricultural sciences is relatively constrained compared with their extensive deployment in other scientific domains [20]. However, as explained by some researchers, they have demonstrated remarkable success in various areas of plant science, including plant breeding [21]. Artificial neural networks (ANNs) represent a category of nonlinear computational techniques employed for various purposes, including grouping data, making predictions, and categorizing intricate systems [22,23]. ANNs can uncover the connections between output and input variables and the underlying insights within datasets without relying on prior physical assumptions or considerations [24]. ANN has played a significant role in various plant sciences, including in vitro germination, regeneration studies, in vitro mutagenesis, and plant system biology [24–29]. This study employed four distinct machine learning models—multilayer perceptron (MLP), radial basis function (RBF), Gaussian process (GP), and extreme gradient boosting (XGBoost)—each with its unique strengths and capacity to capture complex relationships within the data. MLP uses a supervised training process in which the input and output variables are provided as part of the training set. RBF uses the Euclidean distance between each neuron's center and the input as the main input to the neuron's transfer function. GP calculates the likelihood that the input samples belong to a specific class and functions as a nonparametric classifier for binary datasets. Its main advantage is that it works effectively with small datasets, simultaneously providing consistency, precision, and ease of calculation [20]. XGBoost is adept at learning from errors and progressively decreasing the error rate over multiple rounds [27]. The combined use of these models reflects a deliberate effort to utilize a diverse set of machine learning techniques, enhancing the ability of the study to understand the intricate relationships in the dataset involving lavender genotypes, micropropagation, and rooting efficiency.

This study is strategically positioned to contribute to the global interest in lavender cultivation by advancing micropropagation techniques. The overarching goal is to elevate the quality of propagated plants and advocate for sustainable cultivation practices. This study pursued multifaceted objectives, including exploring diverse lavender genotypes, optimizing culture media components, assessing plant growth regulators, and successfully acclimatizing micropropagated plants to external conditions. This study aims to integrate artificial neural network (ANN) analysis and machine learning approaches to enhance the research scope. Computational techniques have been used to model and predict the effects of various culture media components and plant growth regulators on micropropagation quality. Additionally, they analyze and optimize lavender genotype characteristics, ultimately improving the successful acclimatization of micropropagated plants to external

conditions. These advanced tools enhance our understanding and augment our predictive capabilities in lavender cultivation. The study aspires to contribute to sustainable and productive farming methods in lavender cultivation through this comprehensive approach. By achieving these goals, this research aims to provide valuable insights and practical recommendations for lavender growers and enthusiasts seeking to meet the demands of the lavender industry while conserving valuable genetic resources and ensuring the long-term sustainability of lavender cultivation.

## 2. Materials and Methods

### 2.1. Plant Material

Derived from a preceding thesis study conducted by Dalda Şekerci [6], the current research harnessed the potential of four discrete lavender genotypes (160, 175, 183, and 198), distinguished by their exceptional attributes of plant characteristics, essential oil content, and antioxidant capacity. These plant materials were generated through seed segregation populations, a methodology meticulously detailed by Dalda Şekerci [6] in the study mentioned above.

### 2.2. Plant Tissue Culture Works

2.2.1. Sterilization of Shoot Tips

Before being introduced into the culture, the shoot tips were thoroughly sterilized. Initially, the shoot tips were rinsed with tap water for ten minutes. Subsequently, they were immersed in a 70% ethyl alcohol solution for four minutes and then submerged in a 10% sodium hypochlorite solution for two minutes. The shoot tips were meticulously rinsed three times with sterile distilled water within a sterile laminar flow cabinet to eliminate residual sterilizing agents.

2.2.2. Micropropagation

Following the sterilization procedure, the shoot tips were placed into an MS [30] nutrient medium supplemented with various concentrations of BAP (0, 0.5, 1, and 2 mg/L), in addition to 8 g/L agar. All media were repeated with and without 2 g/L AC. The plants were maintained under controlled conditions, including a temperature of $25 \pm 2$ °C and photoperiod of 16/8 h (light/dark). Subculturing of the plants was performed at four-week intervals, with three subcultures being conducted throughout the experiments.

2.2.3. In Vitro Rooting

In vitro rooting trials were conducted using MS nutrient media supplemented with varying concentrations of IBA (0, 0.5, 1, and 2 mg/L), along with 8 g/L agar. All media were repeated with and without 2 g/L AC. The plants were maintained under controlled conditions at a constant temperature of $25 \pm 2$ °C and a photoperiod of 16/8 h (light/dark). The plants remained in the solid media for four weeks.

### 2.3. Acclimatization of Plants to External Conditions

Before transferring the rooted plants to external conditions, the lids of the culture plates were progressively opened to facilitate a pre-acclimatization phase within the laboratory. Subsequently, the plants were transplanted into vials filled with a sterile mixture of peat and perlite at a 1:1 ratio within a controlled greenhouse environment. The plantlets derived from various lavender genotypes and culture media were carefully monitored in separate vials under the shelter of a small mini tunnel to ensure a gradual adjustment to external conditions.

### 2.4. Experimental Design, Statistical Analyses, and Investigated Characteristics

The tissue culture experiments were executed following a factorial design with three replications. Fifteen shoot tips were used for each lavender genotype within the solid culture. In the context of micropropagation, parameters such as multiplication coefficient

and plant height (in centimeters) were recorded. In the in vitro rooting trials, variables encompassing rooting ratio (%), number of roots, and root length were quantified four weeks after culture initiation. Statistical analysis of the experimental data was conducted using JMP 5.01. Percentage (%) values underwent arcsine transformation to meet the assumptions of the statistical tests. Significance among means was determined through the least significant difference (LSD) test, with multiple comparison tests applied at a 5% significance level.

### 2.5. Modeling Procedure

In this study, two well-established artificial neural networks (ANNs) (multilayer perceptron (MLP) and radial basis function (RBF)) and two machine learning (ML) algorithms (Gaussian process (GP) and extreme gradient boosting (XGBoost)) were utilized for the modeling and prediction of micropropagation and rooting efficiency of diverse lavender genotypes through activated carbon and plant growth regulators. The dataset was divided into training and testing subsets with a five-fold cross-validation approach with 20 repetitions to rigorously assess the predictive performance of the ANNs. However, no repetitions were used in the ML algorithms (XGBoost and GP).

The input variables encompassed four distinct genotypes, and 2 g/L AC and four different concentrations of BAP were used as input variables. In contrast, the target (output) variables comprised micropropagation rate, plant height, number of roots, and root length. R programming language was used for coding with the aid of the Caret and Kernlab packages.

The evaluation and comparison of the effectiveness and precision of the ANNs were executed through the utilization of various metrics, including the coefficient of determination ($R^2$), which estimates the strength of the relationship between the model and dependent variable; root mean square error ($RMSE$), an indicator of how closely the regression line aligns with the observed data points; and mean absolute error ($MAE$), which measures the average amount of error between the observed and predicted values (Equations (1)–(3)). These metrics offer a comprehensive assessment of the performance of ANNs in modeling and predicting the micropropagation and rooting efficiency of diverse lavender genotypes through activated carbon and plant growth regulators.

$$R^2 = 1 - \frac{\sum_{i=1}^{n}(Y_i - \hat{Y}_i)^2}{\sum_{i=1}^{n}(Y_i - \tilde{Y})^2} \tag{1}$$

$$RMSE = \sqrt{\frac{\left(\sum_{i=1}^{n}(Y_i - \hat{Y}_i)^2\right)}{n}} \tag{2}$$

$$MAE = \frac{1}{n}\sum_{i=1}^{n}|Y_i - \hat{Y}_i| \tag{3}$$

While $Y_i$ = actual value and $\hat{Y}_i$ = predicted value, $\tilde{Y}$ = mean of the actual values and $n$ = sample count.

#### 2.5.1. Multilayer Perceptron (MLP) Model

The multilayer perceptron (MLP), recognized as one of the foremost artificial neural networks (ANNs), is structured with multiple layers, comprising at least one hidden layer in addition to an input layer and an output layer, as depicted in Figure 1a. MLP employs a

supervised training procedure in which it is presented with both input and output variables as part of the training set. The training process persists until Equation (4) is minimized.

$$E = \frac{1}{n} \sum_{n=1}^{n} (y_s - \hat{y_s}) \tag{4}$$

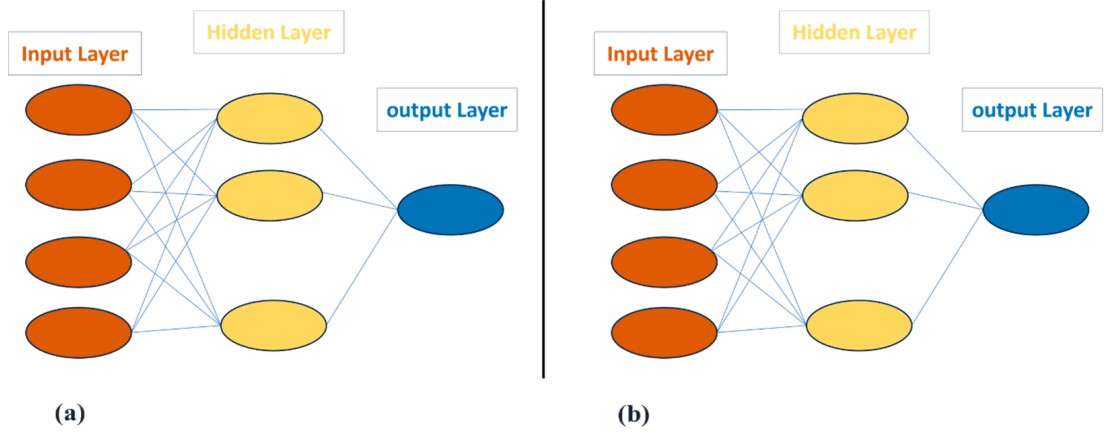

**Figure 1.** ANN models; (**a**) multilayer perceptron, (**b**) radial basis function.

In Equation (4), $n$ is the number of observations, $ys$ is the $s$th observation variable, and $\hat{y}_s$ is the $s$th of predicted variable.

To calculate the predicted value $\hat{y}$ in the multilayer perceptron (MLP), which has a hidden layer with p neurons and k output variables, the following equation is applied.

$$\hat{y} = f \left[ \sum_{j=1}^{p} w_{ji} \cdot g \left( \sum_{i=1}^{k} w_{ji} x_i + w_{j0} \right) + w_o \right] \tag{5}$$

In Equation (5), $x_i$ represents the $i$th output variable, $w_j$ corresponds to the weighted input data entering the $j$th hidden neuron, $f$ is the activation function applied to the output neuron, $w_{ji}$ signifies the weight associated with the direct connection from input neuron $i$ to hidden neuron $j$, $w_{j0}$ represents the bias specific to the $j$th neuron, $w_0$ represents the bias linked to the output neuron, and $g$ is the activation function utilized for the hidden neuron.

2.5.2. Radial Basis Function

The radial basis function (RBF) network is a three-layer artificial neural network comprising an input layer, hidden layer, and output layer (Figure 1b). It uses the Euclidean distance between each neuron's center and input as the primary input to the transfer function of the neuron. The equations that define the Gaussian function, a well-known transfer function in RBF, are as follows:

$$f = (X_r, X_b) = e^{-[||X_r - X_b|| \times 0.8326/h]^2} \tag{6}$$

$X_r$, $X_b$, and $h$ are input with unknown output, observed inputs in time $b$, and spread, respectively.

The dependent variable ($Y_r$) by predictor $X_r$ is calculated as follows:

$$Y_r = \sum_{b=1}^{m} w_b \times f(X_r, X_b) + w_0 \tag{7}$$

### 2.5.3. Extreme Gradient Boosting (XGBoost) Model

Chen and Guestrin [31] created the XGBoost algorithm, a potent tool for handling regression and classification issues. It belongs to the gradient boosting decision tree class and is renowned for its outstanding performance and speed. In a gradient boosting framework, XGBoost is particularly good at learning from mistakes and gradually lowering the error rate over several rounds.

$$y_i = F(x_i) = \sum_{(d=1)}^{D} f_d(x_i), \ f_d \in F, \ i = 1, \dots, n \tag{8}$$

$$L_j = \sum_{i=1}^{n} l(y_i, \ \hat{y}^{(j-1)} + f_j(x_i) + \Omega(f_j) \tag{9}$$

The XGBoost iterative model is displayed in Equation (9), while the XGBoost objective function is indicated in Equation (8).

### 2.5.4. Gaussian Process (GP) Model

The supervised learning GP model extends the Gaussian probability distribution to explain random variable spread, which helps with classification and regression problems. It computes the likelihood that input samples will fall into a certain class and functions as a nonparametric classifier for binary datasets [32]. Its main benefit is that it works well with tiny datasets, offering consistency, precision, and ease of calculation simultaneously [33]. The derivation of the procedure for each input ($x$) and matching output ($y$) is shown in Equation (10).

$$y_i = f(x_i) + \varepsilon \tag{10}$$

## 3. Results

### 3.1. Micropropagation

Shoot tips of four different lavender genotypes (obtained through seed segregation populations, a methodology meticulously detailed by Dalda Şekerci [6]) were cultured in growth media with 0.5, 1, and 2 mg/L BAP, both with and without activated charcoal (AC). The multiplication coefficient and plant height (cm) of the resulting plantlets were measured, and the results are shown in Figure 2.

Figure 2 shows the multiplication rate of genotype 160 for each medium condition with respect to the multiplication coefficient and plant height. It demonstrates how quickly the genotype spreads. While lower values indicate a slower rate of multiplication, higher values indicate a faster rate. Genotype 160 was cultivated at a rate of 2.78 in a growth medium containing 1 mg/L BAP without AC, resulting in the maximum multiplication rate. Additionally, 0.5 mg/L BAP without AC demonstrated a high rate of 2.71 multiplication. The lowest multiplication rate, on the other hand, was 1.42 under 0.5 mg/L BAP with 2 g/L AC. The best results in terms of plant height were obtained with 1 mg/L BAP and 2 g/L AC, yielding 4.57 and 4.42, respectively. The media displayed the lowest 2.60 sans BAP and AC (Figure 2).

Our research concentrated on genotype 175 and evaluated its plant height and rate of multiplication in various growth environments (Figure 3). The genotype multiplied at a rate of 2.57 when grown in a medium containing 1 mg/L BAP without AC. Under 1 mg/L BAP without AC, a remarkable multiplication rate of 2.53 was also noted. In contrast, the absence of BAP and AC resulted in the lowest multiplication rate (1.28), indicating a less favorable environment for growth (Figure 3). The most notable plant heights, which reached 4.00, were attained under circumstances with 0 mg/L BAP and 2 g/L AC. On the other hand, the lowest plant height of 2.35 was found in media devoid of both BAP and AC, indicating that the absence of these growth-promoting elements led to a decrease in plant height (Figure 3).

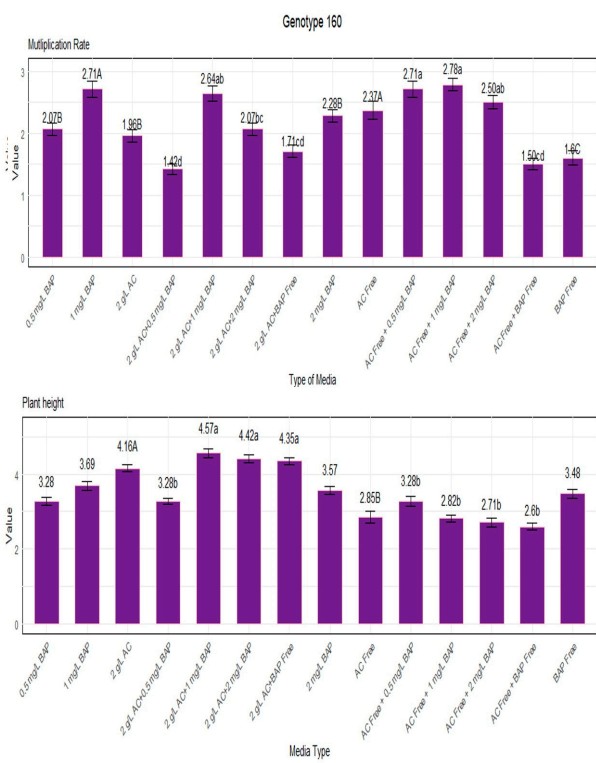

**Figure 2.** Multiplication rate and plant height for genotype 160. Lowercase letters display the interactions between BAP and AC, uppercase letters display the averages of BAP and AC separately.

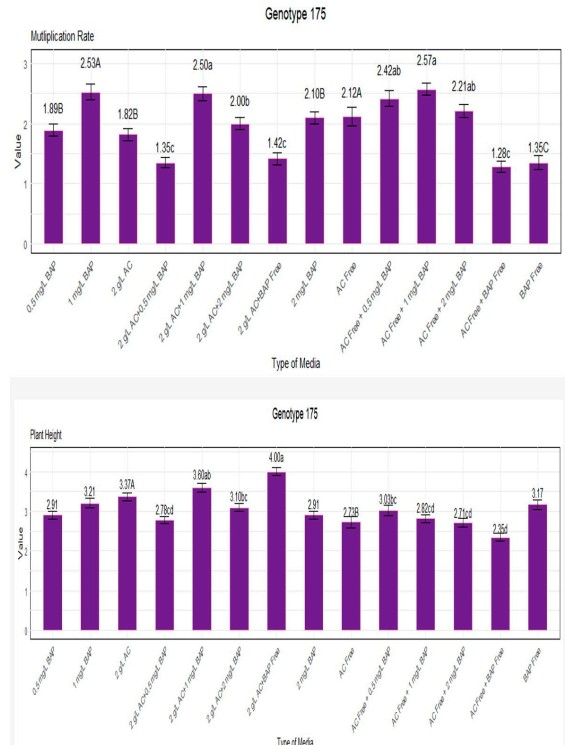

**Figure 3.** Multiplication rate and plant height for genotype 175. Lowercase letters display the interactions between BAP and AC, uppercase letters display the averages of BAP and AC separately.

Figure 4 shows the results of our investigation into the multiplication rate and plant height of genotype 183. When grown in a medium with 1 mg/L BAP and 2 g/L AC, the

genotype showed the most significant multiplication rate of 4.21, as shown in Figure 4. The settings without BAP and with 2 g/L AC had the lowest multiplication rate, 1.42, indicating an unfavorable growing environment.

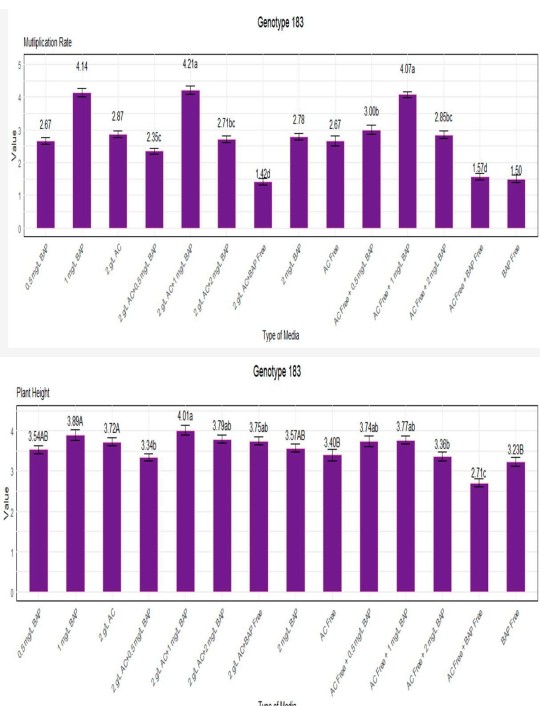

**Figure 4.** Multiplication rate and plant height for genotype 183. Lowercase letters display the interactions between BAP and AC, uppercase letters display the averages of BAP and AC separately.

Continuing to Figure 4, the conditions combining 1 mg/L BAP and 2 g/L AC resulted in a plant height of 4.01, the most significant for genotype 183. In contrast, the media lacking BAP and AC resulted in the lowest plant height, 2.71 inches. These observations clarify the growth traits of genotype 183 under various growing settings.

We measured plant height and multiplication rate of genotype 198 under various growth conditions. Notably, genotype 198 was cultured in a growth medium containing 1 mg/L BAP, both with and without AC, and the maximum multiplication rate of 4.21 was obtained. In contrast, genotype 198 was cultured in media containing 0 mg/L BAP and 0 g/L AC, which resulted in the lowest multiplication rate of 1.17, indicating a less favorable environment for growth (Figure 5).

The best results in terms of plant height were obtained under circumstances with the mixture of 1 mg/L BAP and 2 g/L AC, resulting in a plant height of 4.50. The lowest plant height, 2.60, was observed in medium conditions lacking BAP and AC (Figure 5).

### 3.2. In Vitro Rooting

In vitro rooting tests on lavender genotype 160 produced the best rooting ratio (85.71%) when 1 mg/L IBA was combined with 2 g/L AC. Meanwhile, 0.5 g/L achieved without AC had the lowest value (42%) (Table 1).

**Table 1.** Rooting percentage (%) of genotype 160.

| AC | IBA Free | 0.5 mg/L IBA | 1 mg/L IBA | 2 mg/L BAP | AC Average |
|---|---|---|---|---|---|
| 2 g/L AC | 57.14 (51.42) | 71.42 (64.28) | 85.71 (77.14) | 71.42 (64.28) | 71.42 (64.28) |
| AC free | 57.14 (51.42) | 42.85 (38.57) | 57.14 (51.42) | 71.42 (64.28) | 51.14 (51.42) |
| IBA average | 57.14 (51.42) | 57.14 (51.42) | 71.42 (64.28) | 71.42 (64.28) | |

$LSD_{AC}$: N.S., $LSD_{IBA}$: N.S., $LSD_{AC*IBA}$: N.S., N.S., not significant. Percentages were subjected to angle transformation, and these values are presented in parentheses.

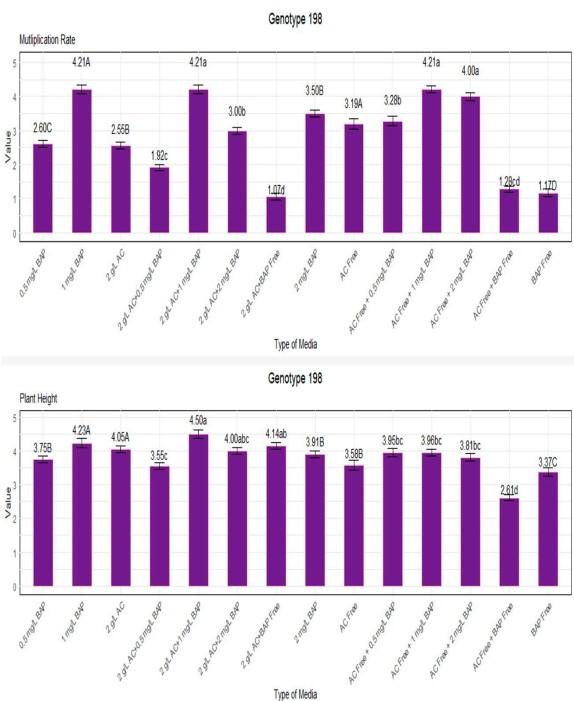

**Figure 5.** Multiplication rate and plant height for genotype 198. Lowercase letters display the interactions between BAP and AC, uppercase letters display the averages of BAP and AC separately.

A thorough investigation was performed using the table that shows genotype 160's root length and root count under various IBA (indole-3-butyric acid) and AC (active carbon) conditions. The growth medium containing 1 mg/L IBA and 2 g/L AC had the most significant average root number (2.28). However, when the growing medium contained 0.5 mg/L IBA without AC, the lowest average root number, 0.92, was discovered.

When the growth medium contained 1 mg/L IBA and 2 g/L AC, the root length average was the greatest, reaching 3.12. On the other hand, the lowest average root length, 1.17, was found in circumstances with 0.5 mg/L IBA and no AC. These findings highlight how sensitive root development is to different IBA concentrations and the presence or absence of AC, and they provide essential information for maximizing root characteristics in the study setting (Table 2).

**Table 2.** Number of roots and root lengths (cm) of genotype 160.

| AC | IBA Free | | 0.5 mg/L IBA | | 1 mg/L IBA | | 2 mg/L BAP | | AC Average | |
|---|---|---|---|---|---|---|---|---|---|---|
| | No. of Roots | Root Length | No. of Roots | Root Length | No. of Roots | Root Length | No. of Roots | Root Length | No. of Roots | Root Length |
| 2 g/L AC | 1.28 | 1.64 | 1.64 | 2.15 | 2.28 | 3.12 | 1.64 | 2.47 | 1.71 | 2.35A |
| AC free | 1.42 | 1.82 | 0.92 | 1.17 | 1.42 | 1.53 | 1.35 | 2.17 | 1.28 | 1.67B |
| IBA average | 1.35 | 1.73 | 1.28 | 1.66 | 1.85 | 2.32 | 1.50 | 2.32 | | |

No. of roots: LSD$_{AC}$: N.S., LSD$_{IBA}$: N.S., LSD$_{AC*IBA}$: N.S.; root length: LSD$_{AC}$:0.55, LSD$_{IBA}$: N.S., LSD$_{AC*IBA}$: N.S.; N.S., not significant.

During rooting tests on lavender genotype 175 plants, it was found that exposure to growth media containing 1 mg/L and 2 mg/L IBA without AC resulted in the highest rooting ratio (85.71%) (Table 3). On the other hand, when genotype 175 was exposed to a condition of media without IBA and AC, the least advantageous rooting ratio (32.14%) was observed, indicating a less suitable environment for root production.

When the growing medium contained 1 mg/L IBA and 2 g/L AC, the genotype 175's average root number, which stood at 2.14, was the greatest. The growth medium with 0 mg/L IBA and 2 g/L AC had the lowest average root number, measuring 0.50. The most significant average was attained in the root length when the growth medium contained

1 mg/L IBA and 2 g/L AC, reaching 2.64. In contrast, the conditions with 0 mg/L IBA and 2 g/L AC had the lowest average root length, 0.46. These data also demonstrate root development's sensitivity to different IBA concentrations and the presence or absence of AC, providing vital information for improving root features in the study setting (Table 4).

**Table 3.** Rooting percentage (%) of genotype 175.

| AC | IBA Free | 0.5 mg/L IBA | 1 mg/L IBA | 2 mg/L BAP | AC Average |
|---|---|---|---|---|---|
| 2 g/L AC | 28.57 (25.71) | 42.85 (38.57) | 78.57 (70.71) | 50.00 (45.00) | 50.00 (45.00) |
| AC free | 35.71 (32.14) | 64.28 (57.85) | 85.71 (77.14) | 85.71 (77.14) | 67.85 (61.07) |
| IBA average | 32.14C (28.92) | 53.57BC (48.21) | 82.14A (73.92) | 67.85AB (61.07) | |

$LSD_{AC}$: N.S., $LSD_{IBA}$: 21.93, $LSD_{AC*IBA}$: N.S.; N.S., not significant. Percentages were subjected to angle transformation, and these values are presented in parentheses.

**Table 4.** Number of roots and root lengths (cm) of genotype 175.

| AC | IBA Free | | 0.5 mg/L IBA | | 1 mg/L IBA | | 2 mg/L BAP | | AC Average | |
|---|---|---|---|---|---|---|---|---|---|---|
| | No. of Roots | Root Length | No. of Roots | Root Length | No. of Roots | Root Length | No. of Roots | Root Length | No. of Roots | Root Length |
| 2 g/L AC | 0.50 | 0.46 | 0.85 | 1.03 | 2.14 | 2.64 | 1.14 | 1.60 | 1.16 | 1.43 |
| AC free | 0.57 | 0.85 | 1.28 | 1.85 | 1.85 | 2.41 | 1.78 | 2.54 | 1.37 | 1.91 |
| IBA average | 0.53C | 0.66C | 1.07BC | 1.44B | 2.00A | 2.52A | 1.46AB | 2.07AB | | |

No. of roots: $LSD_{AC}$: N.S., $LSD_{IBA}$: 0.58; root length: $LSD_{AC*IBA}$: N.S., $LSD_{AC}$: N.S., $LSD_{IBA}$:0.70, $LSD_{AC*IBA}$: N.S. N.S., not significant.

In the context of rooting studies performed on lavender genotype 183, it was apparent that the genotype was grown in a growth medium containing 1 mg/L IBA and 2 g/L AC, which resulted in the genotype achieving the maximum rooting ratio (100%) (Table 5). On the other hand, genotype 175, subjected to a condition without IBA or AC in the growth medium, showed the lowest rooting ratio (10.71%), indicating a less favorable environment for root production.

**Table 5.** Rooting percentage (%) of genotype 183.

| AC | IBA Free | 0.5 mg/L IBA | 1 mg/L IBA | 2 mg/L BAP | AC Average |
|---|---|---|---|---|---|
| 2 g/L AC | 14.28(19.28) | 64.28(55.38) | 100 (90.00) | 78.57(70.71) | 64.28A (59.46) |
| AC free | 7.14(−0.00) | 57.14(51.38) | 85.71(77.14) | 69.23(57.85) | 54.80B (45.98) |
| IBA average | 10.71C (9.64) | 60.71B (54.64) | 92.85A (83.57) | 73.90B (63.05) | |

$LSD_{AC}$: 13.42, $LSD_{IBA}$:18.88, $LSD_{AC*IBA}$: N.S. N.S., not significant. Percentages were subjected to angle transformation, and these values are presented in parentheses.

The highest average root number in genotype 183, 3.50, was found when the growing medium contained 1 mg/L IBA, as shown in Table 6. Instead, when IBA and AC were absent from the growth media, the lowest average root number—measuring 0.50—was observed. Additionally, when 1 mg/L IBA and 2 g/L AC were used as the growth medium, the average root length increased, reaching 4.07. Conversely, the absence of IBA and AC led to the lowest average root length, recorded at 0.21.

**Table 6.** Number of roots and root lengths (cm) of genotype 183.

| AC | IBA Free | | 0.5 mg/L IBA | | 1 mg/L IBA | | 2 mg/L BAP | | AC Average | |
|---|---|---|---|---|---|---|---|---|---|---|
| | No. of Roots | Root Length | No. of Roots | Root Length | No. of Roots | Root Length | No. of Roots | Root Length | No. of Roots | Root Length |
| 2 g/L AC | 0.21 | 0.35 | 1.35 | 1.71 | 3.42 | 4.07 | 1.57 | 2.34 | 1.64 | 2.12 |
| AC free | 0.14 | 0.21 | 1.71 | 1.76 | 2.71 | 2.93 | 1.76 | 2.06 | 1.58 | 1.74 |
| IBA average | 0.17C | 0.28C | 1.53B | 1.73B | 3.07A | 3.50A | 1.67B | 2.20B | | |

No. of roots: $LSD_{AC}$: N.S., $LSD_{IBA}$: 0.58, $LSD_{AC*IBA}$: N.S.; root length: $LSD_{AC}$: N.S., $LSD^{IBA}$: 0.60, $LSD_{AC*IBA}$: N.S.

In trials using the lavender genotype 198, it was discovered that the rooting medium containing 1 mg/L IBA, with or without 2 g/L AC, produced the genotype's best rooting ratio, which reached 100% (Table 7). The genotype 198 conditions, which lacked both IBA and AC in the growth medium, resulted in the least advantageous rooting ratio, 14.28%, indicating a less favorable environment for root development.

**Table 7.** Rooting percentage (%) of genotype 198.

| AC | IBA Free | 0.5 mg/L IBA | 1 mg/L IBA | 2 mg/L BAP | AC Average |
|---|---|---|---|---|---|
| 2 g/L AC | 21.42 (19.28) | 78.57 (70.71) | 100 (90.00) | 85.71 (77.14) | 71.42 (64.28) |
| AC free | 14.28 (12.85) | 78.57 (70.71) | 100 (90.00) | 85.71 (77.14) | 69.64 (62.67) |
| IBA average | 17.85C (16.07) | 78.57B (70.71) | 100AB (90.00) | 85.71AB (77.14) | |

$LSD_{AC}$: N.S., $LSD_{IBA}$: 16.34, $LSD_{AC*IBA}$: N.S. N.S., not significant. Percentages were subjected to angle transformation, and these values are presented in parentheses.

Regarding root length, the circumstances with 1 mg/L IBA and 2 g/L AC resulted in the most significant average, reaching 4.61, emphasizing the beneficial effects of this combination on root growth. The lowest average root length was found at 0.20 when both IBA and AC were absent, highlighting root development's difficulties in such circumstances (Table 8).

**Table 8.** Number of roots and root lengths (cm) of genotype 198.

| AC | IBA Free | | 0.5 mg/L IBA | | 1 mg/L IBA | | 2 mg/L BAP | | AC Average | |
|---|---|---|---|---|---|---|---|---|---|---|
| | No. of Roots | Root Length | No. of Roots | Root Length | No. of Roots | Root Length | No. of Roots | Root Length | No. of Roots | Root Length |
| 2 g/L AC | 0.50 | 0.83 | 1.92 | 2.35 | 3.92 | 4.61 | 2.00 | 2.67 | 2.08 | 2.61 |
| AC free | 0.21 | 0.20 | 2.28 | 2.88 | 3.85 | 4.11 | 2.14 | 2.68 | 2.12 | 2.47 |
| IBA average | 0.35C | 0.52C | 2.10B | 2.62B | 3.98A | 4.36A | 2.07B | 2.67B | | |

No. of roots: $LSD_{AC}$: N.S., $LSD_{IBA}$: 0.59, $LSD_{AC*IBA}$: N.S., Root length: $LSD_{AC}$: N.S., $LSD_{IBA}$: 0.67, $LSD_{AC*IBA}$: N.S. N.S., not significant.

The effect of different mediums, both with and without AC, on rooting is illustrated in Figure 6.

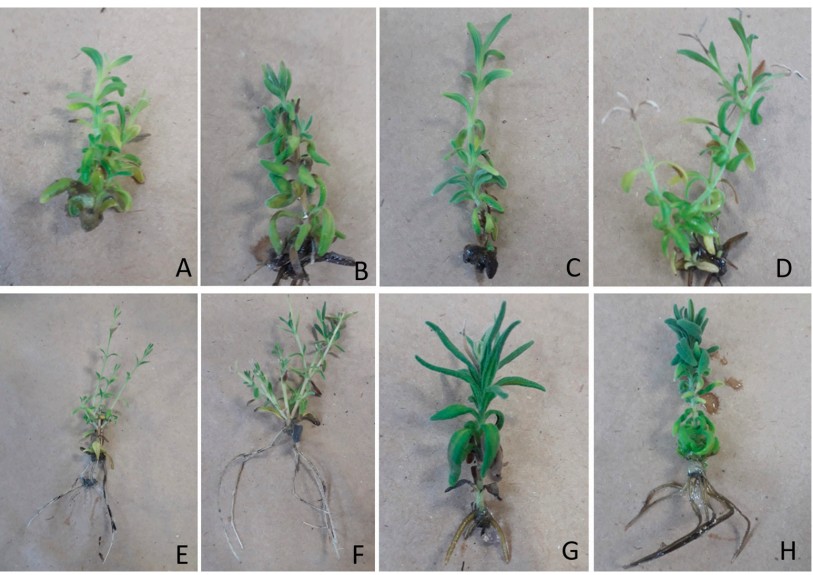

**Figure 6.** (**A**): 160 MS + 1 mg/L BAP, (**B**): 175 MS + 0.5 mg/L BAP + 2 g/L AC, (**C**): 183 MS+ 2 g/L AC, (**D**): 198 MS + 1 mg/L BAP+ 2 g/L AC, (**E**): 183 MS + 1 mg/L IBA + 2 g/L AC, (**F**): 198 MS + 1 mg/L IBA, (**G**): 175 MS + 0.5 mg/L IBA, (**H**): 160 MS + 2 mg/L IBA.

### 3.3. Machine Learning and Artificial Neural Network Analysis

Enhancing the rooting effectiveness and micropropagation of several lavender genotypes is essential. Applying contemporary techniques, such as artificial neural network (ANN) analyses and machine learning (ML), has provided encouraging opportunities for improving agricultural methods. Using these cutting-edge computational tools to predict, optimize, and fine-tune the desired goals with fewer errors and experimental efforts is possible.

Our findings were exposed to ANN analysis utilizing the MLP, RBF, and ML algorithms using XGBoost and GP. The results were validated and forecast using three performance measures ($RMSE$, $R^2$, and $MAE$).

In the comparative evaluation of different machine learning models and their performance metrics for various plant characteristics, the $R^2$ values in the MLP model ranged from 0.24 to 0.85. The highest $R^2$ was observed for root length ($r^2$ = 0.85), while the lowest was for plant height ($r^2$ = 0.24). The mean absolute error ($MAE$) values were generally low, with the number of roots displaying the lowest $MAE$ (0.41). Root mean square error ($RMSE$) values varied from 0.64 to 1.08, with plant height recording the highest $RMSE$ value and the lowest number of roots. For the RBF model, $R^2$ values ranged from 0.42 to 0.80, with the highest $R^2$ observed for root length ($r^2$ = 0.80) and the lowest for plant height ($r^2$ = 0.42). The $MAE$ values also exhibited generally low levels (0.46 to 0.72), while plant height registered the highest $RMSE$ value, and number of roots achieved the minimum $RMSE$. For the XGBoost model, $R^2$ values ranged from 0.46 to 0.87, with the highest $R^2$ in root length ($r^2$ = 0.87) and the lowest in plant height ($r^2$ = 0.46). The $MAE$ values were generally low (0.40 to 0.75), micropropagation rate exhibited the highest $RMSE$ value, and root length displayed the lowest $RMSE$. For the GP model, $R^2$ values ranged from 0.42 to 0.84, with the highest $R^2$ observed for root length ($r^2$ = 0.84) and the lowest for plant height ($r^2$ = 0.42). The $MAE$ values for the GP model were generally low (0.43 to 0.77), and the highest $RMSE$ value was attributed to micropropagation rate. At the same time, the number of roots revealed the minimum $RMSE$ (Table 9).

**Table 9.** Assessment metrics for the ML and ANN models.

|  | Models | *RMSE* | $R^2$ | *MAE* |
|---|---|---|---|---|
| | MLP | 1.08 | 0.24 | 0.88 |
| | RBF | 0.88 | 0.42 | 0.72 |
| Plant height | XGBoost | 0.84 | 0.46 | 0.66 |
| | GP | 0.87 | 0.42 | 0.70 |
| | MLP | 0.92 | 0.57 | 0.71 |
| | RBF | 0.98 | 0.55 | 0.70 |
| Micropropagation rate | XGBoost | 0.98 | 0.51 | 0.75 |
| | GP | 1.00 | 0.48 | 0.77 |
| | MLP | 0.64 | 0.81 | 0.41 |
| | RBF | 0.68 | 0.79 | 0.46 |
| Number of roots | XGBoost | 0.65 | 0.80 | 0.43 |
| | GP | 0.66 | 0.80 | 0.43 |
| | MLP | 0.67 | 0.85 | 0.45 |
| | RBF | 0.78 | 0.80 | 0.54 |
| Root length | XGBoost | 0.63 | 0.87 | 0.40 |
| | GP | 0.74 | 0.84 | 0.60 |

ANN: artificial neural network; ML: machine learning; MLP: multilayer perceptron; RBF: radial basis function; XGBoost: extreme gradient boost; GP: Gaussian process, $R^2$: coefficient of determination; *MAE*: mean absolute error; *RMSE*: root mean square.

Figures 7–10 illustrate the distribution of actual and predicted values for each model, capturing the analysis conducted on various plant metrics. These findings contribute to understanding the predictive capacities and respective performances of the employed models in assessing different plant characteristics.

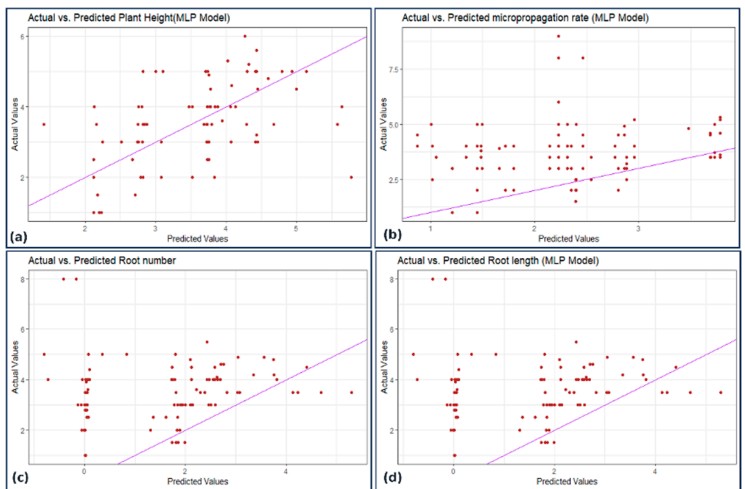

**Figure 7.** Scatterplot actual against predicted values of (**a**) plant height, (**b**) micropropagation rate, (**c**) number of roots, and (**d**) root length using MLP model analysis.

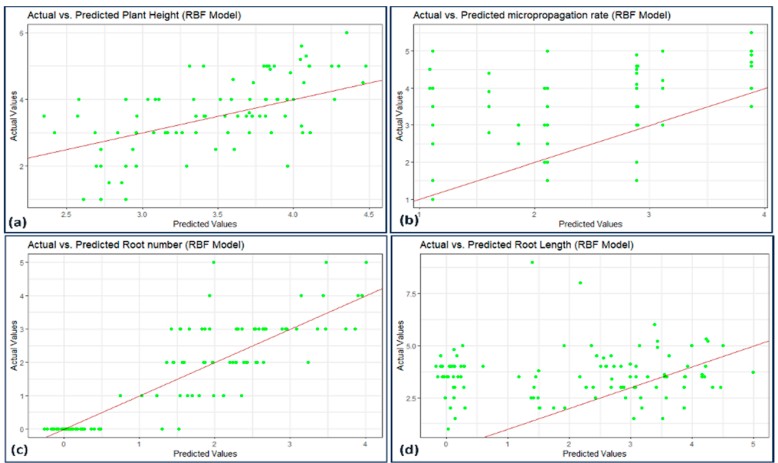

**Figure 8.** Scatterplot actual against predicted values of (**a**) plant height, (**b**) micropropagation rate, (**c**) number of roots, and (**d**) root length using BRF model analysis.

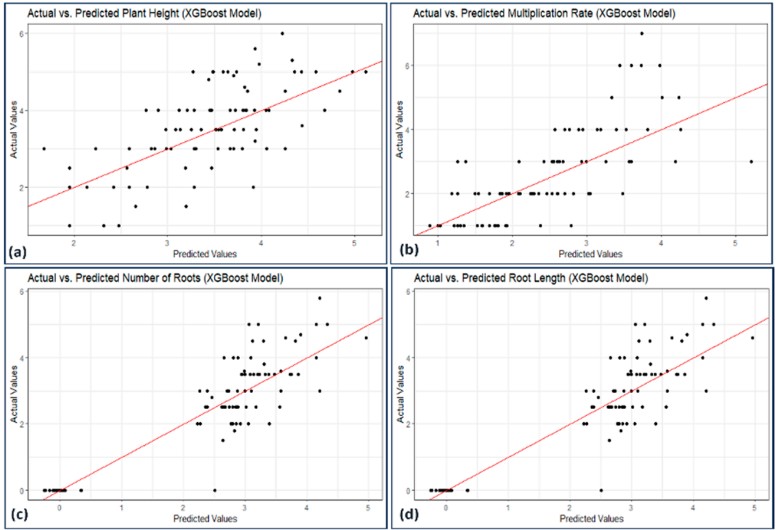

**Figure 9.** Scatterplot actual against predicted values of (**a**) plant height, (**b**) micropropagation rate, (**c**) number of roots, and (**d**) root length using XGBoost model analysis.

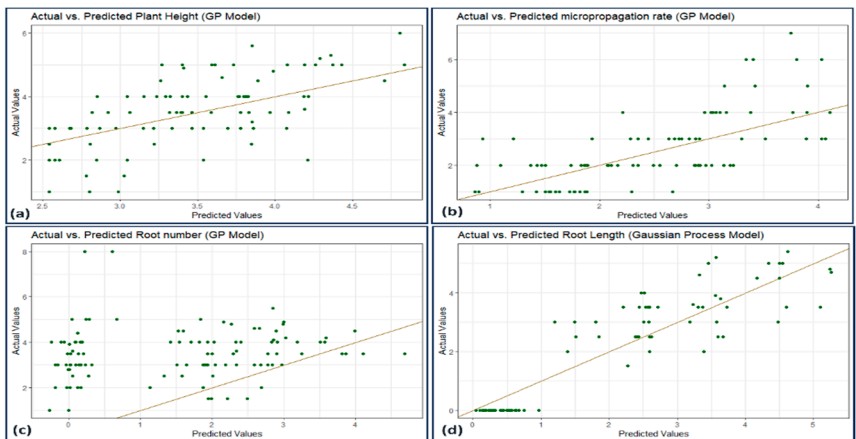

**Figure 10.** Scatterplot actual against predicted values of (**a**) plant height, (**b**) micropropagation rate, (**c**) number of roots, and (**d**) root length using GP model analysis.

The ML and ANN algorithms' $R^2$ rates for plant height were noted in the following order: XGBoost > RBF = GP > MLP. The $R^2$ rates of micropropagation were also MLP > RBF > XGBoost > GP, and number of roots were also ordered as MLP > XGBoost = GP > BRF. Lastly, the $R^2$ rate of root length was ordered as XGBoost > MLP > GP > BRF.

In conclusion, XGBoost seems to outperform the other models, as indicated by the findings. It performed exceptionally well in predicting root length and showed higher $R^2$ values, reduced *MAE*, and competitive *RMSE* outcomes across all assessed plant parameters.

## 4. Discussion

Modern biotechnological methods, including tissue culture techniques, play a crucial role in enhancing the outcomes of plant breeding efforts. Tissue culture techniques have evolved into practical tools for developing new cultivars [34]. Among these techniques, micropropagation has gained significant prominence for its ability to propagate and induce root formation in various plant species rapidly. Tissue culture techniques involve cultivating plant cells, tissues, or organs in a controlled environment, allowing for precise control over the growth and development of plants. This technology has opened up new possibilities for plant breeders to produce superior and genetically uniform plant varieties. Micropropagation has become a go-to method for clonal propagation, enabling the mass production of genetically identical plantlets from a single parent plant. This method offers advantages such as speed, efficiency, and the ability to propagate plants with desirable traits, such as disease resistance, high yield, or unique characteristics. Furthermore, tissue culture techniques have revolutionized the rooting process, enabling the formation of roots in plant cuttings under controlled conditions. This has accelerated the production of healthy and well-established plants for various agricultural and horticultural purposes [35].

The successful micropropagation of lavender plants hinges on several crucial factors, including the optimization of culture conditions and the strategic use of growth regulators. For instance, the incorporation of benzylaminopurine and $\alpha$-naphthaleneacetic acid in the culture medium has been demonstrated to enhance the production of multiple shoots from nodal segment explants, as highlighted in a study by Frabetti et al. [36]. Furthermore, fine-tuning light conditions, such as employing red filters and eliminating indolebutyric acid from the growth medium, has been shown to improve the in vitro rooting of lavender plants, as discussed in research by Rodrigues et al. [37].

In our study, cultivation conditions, specifically the concentrations of benzylaminopurine (BAP) and activated charcoal (AC), significantly affected the multiplication rate and plant height of each genotype. For genotype 160, the highest multiplication rate (2.78) was achieved under conditions with 1 mg/L BAP without AC, while 0.5 mg/L BAP without AC also demonstrated a notable rate of 2.71. The lowest multiplication rate (1.42) occurred under conditions with 0.5 mg/L BAP and 2 g/L AC. Regarding plant height, the optimal

results were observed with 1 mg/L BAP and 2 g/L AC, yielding heights of 4.57 and 4.42, respectively. The absence of both BAP and AC resulted in the lowest plant height (2.60). Genotype 175 exhibited a multiplication rate of 2.57 under 1 mg/L BAP without AC and a notable rate of 2.53 under the same BAP concentration without AC. The absence of both BAP and AC led to the lowest multiplication rate (1.28). Plant height reached its maximum (4.00) under conditions with 0 mg/L BAP and 2 g/L AC, while the lowest height (2.35) was observed in the absence of both BAP and AC. In the case of genotype 183, the highest multiplication rate (4.21) was achieved when cultivated with 1 mg/L BAP and 2 g/L AC.

Conversely, the absence of BAP and the presence of 2 g/L AC resulted in the lowest multiplication rate (1.42). Plant height followed a similar trend, with the most significant height (4.21) observed under conditions with 1 mg/L BAP and 2 g/L AC, and the lowest height (2.60) in the absence of both BAP and AC. Finally, for genotype 198, the maximum multiplication rate (4.21) was attained with 1 mg/L BAP, with and without AC. The lowest multiplication rate (1.17) was recorded when cultured in media lacking both BAP and AC. Plant height exhibited a similar pattern, with the best results (4.50) under conditions with a combination of 1 mg/L BAP and 2 g/L AC, and the lowest height (2.60) in the absence of both BAP and AC.

These findings underscore the significance of BAP and AC concentrations in the growth medium in influencing the multiplication rate and plant height of the evaluated genotypes. These results suggest that a balanced combination of BAP and AC is crucial for optimizing the growth conditions and overall performance of the studied genotypes.

The findings of our study highlight the significance of leveraging machine learning (ML) techniques, specifically artificial neural network (ANN) analyses and ML algorithms such as XGBoost and genetic programming (GP), to enhance the understanding and optimization of in vitro rooting and micropropagation processes in lavender genotypes. Our results demonstrated the effectiveness of these computational tools in predicting and optimizing critical parameters related to rooting efficiency and micropropagation success. Unlike traditional methods, ML models offer a more efficient and accurate way of predicting outcomes in the complex and nonlinear biological processes involved in in vitro rooting.

Using MLP and RBF algorithms in ANN analysis and XGBoost and GP in ML algorithms allowed the prediction and optimization of various plant characteristics, including root length, plant height, micropropagation rate, and number of roots. The $R^2$ values obtained from the different ML models demonstrated their ability to accurately predict the studied parameters. XGBoost consistently demonstrated superior performance, exhibiting higher $R^2$ values across most plant characteristics, such as root length, micropropagation rate, and the number of roots.

The comparative evaluation of $R^2$ values indicated that XGBoost outperformed other models, followed by MLP, RBF, and GP. The mean absolute error (*MAE*) values, reflecting the accuracy of predictions, were consistently low across all models, affirming the reliability of ML techniques in predicting in vitro rooting responses. Root mean square error (*RMSE*) values further supported the effectiveness of ML models in minimizing prediction errors. In the context of lavender micropropagation, XGBoost emerged as the most robust model, demonstrating its capability to provide accurate predictions with reduced errors. This suggests that XGBoost is well suited for optimizing in vitro parameters in micropropagation studies of lavender genotypes.

The comprehensive analysis of different ML models and their performance metrics strengthens the argument for the practical application of these computational tools in optimizing plant tissue culture protocols for further scientific investigations and biotechnological approaches. In conclusion, our study contributes valuable insights into the application of ML techniques, emphasizing the superiority of XGBoost in predicting and optimizing in vitro parameters for lavender micropropagation. Integrating these advanced computational tools holds promise for streamlining and improving the efficiency of micropropagation protocols in lavender and potentially other plant species. Integrating machine

learning (ML) techniques in plant tissue culture studies has become a pivotal strategy for optimizing complex in vitro processes.

Drawing inspiration from Jafari et al. [38], who employed a hybrid generalized regression neural network (GRNN) and genetic algorithm (GA) to predict in vitro rooting responses in *Passiflora caerulea*, our study extends this approach to lavender genotypes. We successfully predicted key micropropagation parameters using diverse ML algorithms, including XGBoost, MLP, RBF, and GP. This aligns with the findings of Demirel et al. [39] and Aasim et al. [26], where machine learning models, particularly XGBoost, demonstrated superior performance in optimizing tissue culture conditions for black chokeberry and predicting outcomes in common bean regeneration, respectively. Furthermore, the study by Sadat-Hosseini et al. [40] on Persian walnut proliferation, employing MLPNN, KNN, and gene expression programming (GEP), emphasizes the importance of accurate modeling for tissue culture media optimization. In our research, we utilized similar ML techniques (XGBoost, MLP, RBF, GP) for predicting and optimizing in vitro parameters in lavender.

Our findings, particularly the outstanding performance of XGBoost, echo the efficacy demonstrated by GEP in the Persian walnut study [40], supporting the notion that ML techniques significantly contribute to the precision and efficiency of tissue culture protocols. The overarching trend observed in these studies, including ours, underscores the transformative impact of ML in plant tissue culture research. The ability of ML models to navigate intricate, nonlinear biological processes provides researchers with a powerful tool for enhancing the accuracy of predictions and optimizing experimental outcomes. As the field continues to embrace these computational advancements, it is anticipated that ML will play an increasingly integral role in shaping the future of plant tissue culture studies, reducing experimental efforts and advancing our understanding of intricate in vitro processes across diverse plant species.

## 5. Conclusions

The successful establishment of in vitro micropropagation and rooting protocols for lavender genotypes, as evidenced by the findings of this study, represents a crucial advancement in ensuring a robust, year-round, and efficient propagation method for lavender plants. The significance of these results lies in overcoming the limitations imposed by seasonal variations and offering a flexible and reliable means for continuous plant propagation. In addition to these achievements, integrating machine learning (ML) techniques into the optimization process can further enhance the efficiency of in vitro micropropagation and rooting protocols for lavender. Machine learning models, such as XGBoost, MLP, RBF, and GP, have effectively predicted and optimized critical parameters in complex biological processes, as highlighted in contemporary plant tissue culture studies.

By incorporating ML into the fine-tuning process of specific variables, including culture media compositions, growth regulators, and activated carbon utilization, we can leverage predictive models to identify optimal conditions for lavender micropropagation. The adaptability of ML models enables a data-driven approach to refining protocols, ensuring precision in selecting growth conditions tailored to the unique characteristics of lavender genotypes. This synergy between traditional in vitro techniques and machine learning advances our understanding of lavender micropropagation and provides a pathway to achieve higher efficiency and consistency in preserving, breeding, and commercial-scale production of diverse lavender genotypes.

The continuous and uninterrupted supply of healthy plant material facilitated by these optimized protocols aligns with the growing demand for lavender plants in various applications. As future studies delve deeper into the interplay of ML and in vitro techniques, there is potential for uncovering novel insights and strategies to improve the sustainability and efficiency of plant propagation methods. The utilization of machine learning in lavender micropropagation represents a progressive step toward harnessing innovative technologies for conserving and exploiting genetic resources in agricultural and conservation contexts.

**Author Contributions:** Conceptualization, Ö.Ş.; methodology, Ö.Ş., A.D.Ş., M.A.I., F.B., M.T. and D.D.; data curation, Ö.Ş. and M.A.I.; writing—original draft preparation, Ö.Ş.; writing—review and editing, Ö.Ş. and T.İ.; visualization, Ö.Ş. and M.A.I.; supervision, Ö.Ş. All authors have read and agreed to the published version of the manuscript.

**Funding:** This research received no external funding.

**Data Availability Statement:** Data are contained within the article.

**Acknowledgments:** The authors would like to thank Eugene Steele, professional English editor of Erciyes University, for the English language editing of the manuscript. We thank the Office of the Dean for Research at Erciyes University for providing the necessary infrastructure and laboratory facilities at the ArGePark research building.

**Conflicts of Interest:** The authors declare no conflict of interest.

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
