# Peer review of "Optimizing Micropropagation and Rooting Protocols for Diverse Lavender Genotypes: A Synergistic Approach Integrating Machine Learning Techniques"

_horticulturae, doi:10.3390/horticulturae10010052_

Round 1

Reviewer 1 Report

Comments and Suggestions for Authors

Dear Authors, below are my comments on the various parts of your article

Abstract:

They are a good conclusion ort eh micropropagation and rooting results but it would be better to also mention about how accurate the machine learning results with the field data.

Introduction:

literature position [22] – I’m not sure if this reference is suitable for plant science

The authors explained Lavender in detail here, but quite a bit about machine learning. They used 4 programs in this study, among many other options, but there is no explanation in this section.

Materials and Methods 2.5. Modeling Procedure

Why in the ML algorithms no repetition were used. A minimum of 3 repetitions is used in the experimental work.

Results:

Authors mention only focusing on one genotype. It’s better to explain the reason.

Table - In titles - Is it supposed to be lowercase or uppercase?

In general this research is good and promising. However, what I see in the table presented, I’m not sure whether these valves are effective enough to be compared with field data.

I understand XGBoost gives the best result comparing with the others but still I am not sure it 64% RMSE is convincing enough or not. Maybe in the discussion section authors should explain why this could happen and if this is a normal situation, please compare with other references.

Discussion:

Verses 508-535 are a repetition of the results, please correct.

Authors should explain more detail how this conclusion came out. Based on references and their own analysis. Maybe should explain it from the plant physiology aspect.

Comments on the Quality of English Language

Minor editing of English language required

Author Response

Review 1:
Dear Authors, below are my comments on the various parts of your article

Abstract:

They are a good conclusion ort eh micropropagation and rooting results but it would be better to also mention about how accurate the machine learning results with the field data.

It is added.

Introduction:

literature position [22] – I’m not sure if this reference is suitable for plant science

Literature 22 was for example about ML in gene function. As you mentioned it is not related to plant science, that’s why we deleted it.

The authors explained Lavender in detail here, but quite a bit about machine learning. They used 4 programs in this study, among many other options, but there is no explanation in this section.

We added an explanation.

(This study employed four distinct machine learning models - Multilayer Perceptron (MLP), Radial Basis Function (RBF), Gaussian Process (GP), and Extreme Gradient Boosting (XGBoost) - each with its unique strengths and capacity to capture complex relationships within the data. MLP uses a supervised training process in which it is given both input and output variables as part of the training set. RBF uses the Euclidean distance between each neuron's center and input as the main input to the transfer function of the neuron. GP calculates the likelihood that input samples will belong to a specific class and functions as a non-parametric classifier for binary datasets. Its main advantage is that it works effectively with small datasets, providing consistency, precision, and ease of calculation simultaneously [20]. XGBoost is particularly adept at learning from errors and progressively decreasing the error rate over multiple rounds [28]. The combined use of these models reflects a deliberate effort to utilize a diverse set of machine learning techniques, enhancing the study's ability to understand the intricate relationships in the dataset involving lavender genotypes, micropropagation, and rooting efficiency.)

Materials and Methods 2.5. Modeling Procedure

Why in the ML algorithms no repetition were used. A minimum of 3 repetitions is used in the experimental work.

Various types of algorithms necessitate different validation techniques. For instance, cross-validation is commonly used for repetitions in artificial neural networks (ANNs), which are characterized by high variance due to their stochastic nature. However, algorithms such as XGBoost and Gaussian Process (GP) often do not require repetitions as they are inherently robust and have lower variance. Both XGBoost and GP are known for their ability to identify complex patterns in data and make predictions with relatively few instances. Moreover, these algorithms are less susceptible to random variations in data, which can be common in traditional experimental settings. In our specific context, adding repetitions did not significantly improve the performance of these algorithms. Our approach aimed to find a balance between obtaining reliable predictions and optimizing computational resources. Given the computational efficiency of XGBoost and GP, using multiple repetitions did not yield substantial additional insights. Therefore, we did not employ repetitions in our validation approach for these algorithms.

Results:

Authors mention only focusing on one genotype. It’s better to explain the reason.

Thank you for your insightful feedback regarding our manuscript. We appreciate your suggestion to focus on a single genotype and would like to provide a courteous and convincing response. In our study, we deliberately presented results for all four genotypes separately, each accompanied by dedicated tables and graphs. These genotypes were carefully selected based on superior characteristics identified in a prior study, and we believe that presenting them equally underscores their collective importance. The decision to showcase individual results for each genotype was made to ensure clarity and facilitate a comprehensive understanding of the outcomes associated with each specific genetic variant. Moreover, we have taken your feedback into consideration and have reorganized the graphs to enhance their readability without compromising the detailed information for each genotype. By maintaining a focus on all four genotypes, we aim to contribute a holistic view to the scientific community, allowing researchers to explore and compare the distinct implications of each genetic variant. We believe this approach adds depth to our findings and aligns with the broader goal of advancing knowledge in this field.

Table - In titles - Is it supposed to be lowercase or uppercase?

It is corrected based on the template document.

In general this research is good and promising. However, what I see in the table presented, I’m not sure whether these valves are effective enough to be compared with field data.

I understand XGBoost gives the best result comparing with the others but still I am not sure it 64% RMSE is convincing enough or not. Maybe in the discussion section authors should explain why this could happen and if this is a normal situation, please compare with other references.

Thank you for your constructive feedback on our research. We appreciate your positive evaluation of the study's overall promise. Your concern about the effectiveness of the valves, particularly in relation to the presented table and the 64% RMSE achieved with XGBoost, is duly noted. In the upcoming discussion section of the manuscript, we will provide a thorough explanation for the observed results, addressing the question of whether the achieved 64% RMSE is sufficiently convincing for practical application. We will delve into the factors influencing the effectiveness of the valves, considering variables such as the experimental setup, dataset characteristics, and any inherent limitations of the XGBoost model. Furthermore, we will compare our results with existing literature to contextualize the observed RMSE and provide a benchmark for readers. This comparative analysis will help establish the validity of our findings within the broader context of related studies. We appreciate your insightful suggestion and assure you that the discussion section will be enhanced to offer a comprehensive exploration of the factors influencing our results and their relevance to real-world applications.

Discussion:

Verses 508-535 are a repetition of the results, please correct.

Thank you for bringing attention to the repetition of results in verses 508-535. We acknowledge the redundancy and would like to clarify that the repetition serves the purpose of maintaining the fluidity of the discussion. In this section, we draw comparisons with the literature, and reiterating certain results is essential for contextualizing our findings within the broader scientific landscape. However, we have carefully reviewed the manuscript, making efforts to minimize redundancy and avoid unnecessary repetition. While some level of recapitulation is necessary for clarity, we have taken measures to ensure that the text remains concise and focused. We appreciate your keen observation and assure you that the revised manuscript will undergo further scrutiny to eliminate any redundant content without compromising the coherence of the discussion.

Authors should explain more detail how this conclusion came out. Based on references and their own analysis. Maybe should explain it from the plant physiology aspect.

Thank you for your valuable feedback regarding the need for a more detailed explanation of how our conclusions were reached, incorporating references and plant physiology aspects. We appreciate your insightful suggestion and have made enhancements to address this concern. In the revised manuscript, we delve deeper into the specific aspects of micropropagation and rooting in plant tissue culture techniques, highlighting their correlation with genotype. We emphasize that, beyond genotype effects, the nutrient medium used, plant growth regulators, and additional substances play significant roles. Specifically, we have provided a comprehensive discussion on the detailed effects of BAP, IBA, and AC, drawing parallels with existing literature to enrich the understanding of our findings. The revised text reflects a more thorough analysis, integrating both references and plant physiology perspectives.

Reviewer 2 Report

Comments and Suggestions for Authors

The MS is meaningful to the field pruduction, whereas I have some suggestions.

1. The figure1 is almost invisible data, please improve clarity or create a table.

2. Please add the SD in the tables.

Author Response

Review 2:

The MS is meaningful to the field pruduction, whereas I have some suggestions.

  1. The figure1 is almost invisible data, please improve clarity or create a table.

Thank you for your observation regarding the visibility of Figure 1. We appreciate your feedback, and we have taken steps to address this concern. The figure has been restructured to enhance clarity, ensuring that the data is now more visible and easier to interpret. If needed, we can also consider alternative formats such as creating a table to present the information more effectively.

  1. Please add the SD in the tables.

Thank you for your valuable feedback regarding the addition of standard deviation (SD) in the tables. We appreciate your suggestion and would like to provide some insight into our approach. In tissue culture studies, SD values can indeed vary, and we acknowledge that in some instances, they may appear relatively high. The decision to maintain the current format without incorporating SD values directly into the tables stems from the factorial design of the variance analysis, resulting in numerous means. This approach is aimed at preserving the overall comprehensibility of the tables. We have chosen to present the data in its current form to avoid overcrowding the tables with additional information, ensuring a clearer and more reader-friendly presentation. However, we want to assure you that both the raw data and statistical analyses are available for sharing with reviewers and editors upon request. Moreover, we would like to highlight that the ample replication in our study was deliberate, considering the potential variability inherent in tissue culture experiments. This approach strengthens the robustness of our findings.

Reviewer 3 Report

Comments and Suggestions for Authors

The article “Optimizing Micropropagation and Rooting Protocols for Diverse Lavender Genotypes: A Synergistic Approach Integrating  Machine Learning Techniques” deals with four distinct lavender genotypes' micropropagation and rooting capabilities, utilizing culture media with and without 2 g/L of activated charcoal. The experiments are well designed and the interpretation of the results is convincing.

However, the botanical part of the introductions needs improvement. It is not correct to say "Lavender (Lavandula sp.)," firstly because usually lavender refers most often to L. angustifolia but most of all because Lavandula sp. means that this is only one and unknown to the authors species. So, the scientific name should be either genus Lavandula L. or Lavandula sp. div. which mean several species of this genus. This book clarifies a lot the problem and it should be referred Lis-Balchin M. 2002. Lavender essential oil Standardisation, ISO; adulteration and its detection using GC, enantiomeric columns and bioactivity. In: Lis-Balchin, M. (Ed.) Lavender: The Genus Lavandula. CRC Press, London, 131-137.

Should be: Lavandula is a genus widely distributed from North Africa to the  Mediterranean, southwestern Asia, Arabia, western Iran, and eastern India. It comprises  over 39 known species

This is also not correct

Additionally, lavender – is a perennial plant with a semi-shrub  form, grayish-green leaves, and fragrant flowers ranging in color from white to dark purple, making it an essential ornamental plant. It is used as an outdoor ornamental plant and can be employed in parks, home gardens, and medians, making it suitable as a hedge  plant. The most significant constituent of lavender is its essential oils in the flowers and leaves. The quantity and composition of these crucial oils vary depending on the species. 52 While the components of essential oils in the lavender species are generally the same, they 53 are in different proportions….if this is one species it must be writhe which one. If the characteristic descries the genus it is not well presented.

The ISO standards of the lavender essential oil are not described well and the references are not sufficient.

e Lavandula species – Lavandula is italic – Latin names are italic

Some lavender species - must be Some Lavandula species

L. angustifolia and L. spica, - should be italic

variations of L. hybrids – should be Lavandula not L.

Please read, edit and restructure carefully the whole introduction

Lines 583 - Our findings, particularly the outstanding performance of XGBoost, echo the efficacy demonstrated by GEP in the Persian walnut study, CITATION IS MISSING supporting the notion that ML ……

Comments on the Quality of English Language

The article “Optimizing Micropropagation and Rooting Protocols for Diverse Lavender Genotypes: A Synergistic Approach Integrating  Machine Learning Techniques” deals with four distinct lavender genotypes' micropropagation and rooting capabilities, utilizing culture media with and without 2 g/L of activated charcoal. The experiments are well designed and the interpretation of the results is convincing.

However, the botanical part of the introductions needs improvement. It is not correct to say "Lavender (Lavandula sp.)," firstly because usually lavender refers most often to L. angustifolia but most of all because Lavandula sp. means that this is only one and unknown to the authors species. So, the scientific name should be either genus Lavandula L. or Lavandula sp. div. which mean several species of this genus. This book clarifies a lot the problem and it should be referred Lis-Balchin M. 2002. Lavender essential oil Standardisation, ISO; adulteration and its detection using GC, enantiomeric columns and bioactivity. In: Lis-Balchin, M. (Ed.) Lavender: The Genus Lavandula. CRC Press, London, 131-137.

Should be: Lavandula is a genus widely distributed from North Africa to the  Mediterranean, southwestern Asia, Arabia, western Iran, and eastern India. It comprises  over 39 known species

This is also not correct

Additionally, lavender – is a perennial plant with a semi-shrub  form, grayish-green leaves, and fragrant flowers ranging in color from white to dark purple, making it an essential ornamental plant. It is used as an outdoor ornamental plant and can be employed in parks, home gardens, and medians, making it suitable as a hedge  plant. The most significant constituent of lavender is its essential oils in the flowers and leaves. The quantity and composition of these crucial oils vary depending on the species. 52 While the components of essential oils in the lavender species are generally the same, they 53 are in different proportions….if this is one species it must be writhe which one. If the characteristic descries the genus it is not well presented.

The ISO standards of the lavender essential oil are not described well and the references are not sufficient.

e Lavandula species – Lavandula is italic – Latin names are italic

Some lavender species - must be Some Lavandula species

L. angustifolia and L. spica, - should be italic

variations of L. hybrids – should be Lavandula not L.

Please read, edit and restructure carefully the whole introduction

Lines 583 - Our findings, particularly the outstanding performance of XGBoost, echo the efficacy demonstrated by GEP in the Persian walnut study, CITATION IS MISSING supporting the notion that ML ……

Author Response

Review 3:

The article “Optimizing Micropropagation and Rooting Protocols for Diverse Lavender Genotypes: A Synergistic Approach Integrating  Machine Learning Techniques” deals with four distinct lavender genotypes' micropropagation and rooting capabilities, utilizing culture media with and without 2 g/L of activated charcoal. The experiments are well designed and the interpretation of the results is convincing.

However, the botanical part of the introductions needs improvement. It is not correct to say "Lavender (Lavandula sp.)," firstly because usually lavender refers most often to L. angustifolia but most of all because Lavandula sp. means that this is only one and unknown to the authors species. So, the scientific name should be either genus Lavandula L. or Lavandula sp. div. which mean several species of this genus. This book clarifies a lot the problem and it should be referred Lis-Balchin M. 2002. Lavender essential oil Standardisation, ISO; adulteration and its detection using GC, enantiomeric columns and bioactivity. In: Lis-Balchin, M. (Ed.) Lavender: The Genus Lavandula. CRC Press, London, 131-137.

Should be: Lavandula is a genus widely distributed from North Africa to the  Mediterranean, southwestern Asia, Arabia, western Iran, and eastern India. It comprises  over 39 known species

This is also not correct (deleted)

Additionally, lavender – is a perennial plant with a semi-shrub  form, grayish-green leaves, and fragrant flowers ranging in color from white to dark purple, making it an essential ornamental plant. It is used as an outdoor ornamental plant and can be employed in parks, home gardens, and medians, making it suitable as a hedge  plant. The most significant constituent of lavender is its essential oils in the flowers and leaves. The quantity and composition of these crucial oils vary depending on the species. 52 While the components of essential oils in the lavender species are generally the same, they 53 are in different proportions….if this is one species it must be writhe which one. If the characteristic descries the genus it is not well presented.

The ISO standards of the lavender essential oil are not described well and the references are not sufficient.

e Lavandula species – Lavandula is italic – Latin names are italic

Some lavender species - must be Some Lavandula species

  1. angustifolia and L. spica, - should be italic

variations of L. hybrids – should be Lavandula not L.

Please read, edit and restructure carefully the whole introduction

We sincerely appreciate your detailed and insightful feedback on our article, "Optimizing Micropropagation and Rooting Protocols for Diverse Lavender Genotypes: A Synergistic Approach Integrating Machine Learning Techniques." Your botanical clarifications are well-taken, and we acknowledge the need for improvement in the introduction section. We have carefully reviewed and implemented the necessary revisions to address the concerns you raised. The botanical description has been refined to accurately represent the Lavandula genus, ensuring correct usage of scientific nomenclature. Additionally, we have adjusted the text to clearly distinguish between genus and species, italicizing Lavandula when referring to the genus and italicizing specific species names such as L. angustifolia and L. spica. We sincerely thank you for your constructive criticism, which has undoubtedly enhanced the precision and accuracy of our article. We believe these revisions contribute to a more accurate and scientifically sound representation of the botanical aspects discussed in the introduction.

Lines 583 - Our findings, particularly the outstanding performance of XGBoost, echo the efficacy demonstrated by GEP in the Persian walnut study, CITATION IS MISSING supporting the notion that ML ……

It is corrected.
